# Pandemic-induced changes in household-level food diversity and diet quality in the U.S.

**Daniel P. Simandjuntak, Edward C. Jaenicke** *, Douglas H. Wrenn

Department of Agricultural Economics, Sociology, and Education, Penn State University, University Park, Pennsylvania, United States of America

* ecj3psu.edu

## Abstract

Using household-level U.S. food-purchase data, this study investigates pandemic-induced changes in two measures of diet quality, a *Berry Index* that measures food diversity and a *USDAScore* that measures adherence to dietary guidelines. Using NielsenIQ consumer panel data and a total of XXX households that neither moved location nor left the panel from 2018 through 2020, we estimate an event-study model where each household's diet quality measures before and during the pandemic period are compared against the same households' scores one year prior. In the two-to-three months following pandemic-based school closures, which spanned the March-June 2020 period, depending on the specific U.S. state, we find modest increases in food diversity (of up to 2.6 percent compared to the prior year) for the *Berry Index* and larger temporary increases (of up to 8.5 percent) in diet quality as measured by the *USDAScore*. We also find that households with different demographic characteristics generally follow the same patters; however, households with young children, low-income households, and households that do not own a vehicle exhibit smaller increases.

**Data Availability Statement:** All relevant details on how researchers may access the third-party data used in this study via subscription with the Kilts Center for Marketing at the University of Chicago (https://www.chicagobooth.edu/research/kilts) are

## Introduction

In addition to presenting a direct risk to human health and creating adverse shocks to health care systems, the global COVID-19 pandemic disrupted food acquisition and consumption in ways that had not been seen in generations. In the U.S., starting in March 2020, the pandemic shut down almost all restaurants and cafeterias [1] and disrupted agricultural supply chains [2] so that retail food shoppers encountered stock-outs of some items for the first time in decades [3]. Since approximately 50 percent of U.S. food dollars are spent on food away from home (FAFH) options [4], the pandemic-induced closures and lockdowns set the stage for a major shift from FAFH establishments to supermarkets and other food retailers that remained open during the pandemic. This shift may have dramatically affected the composition of food acquisitions and overall diet quality.

In the U.S., prior to the pandemic, the average U.S. diet could be described as generally unhealthy and shows only minor signs of improving: For example, the *Dietary Guidelines for Americans* [5] says that "eating patterns in the United States have remained far below Dietary Guidelines recommendations" and documents only a slight improvement in the U.S.

within the paper and its Supporting information files. Five Excel files holding publicly available data used in the paper along with the proprietary data are available from the Harvard Dataverse repository (https://doi.org/10.7910/DVN/S6XAFY). Step-by-step instructions on how to combine datasets, construct variables, and replicate the tables are within the paper and its Supporting information files. The authors confirm that others would be able to access or request these data in the same manner as themselves. The authors also confirm that they did not have any special access or request privileges that others would not have.

**Funding:** All three co-authors (DPS, ECJ, and DHW) received partial funding from Open Philanthropy Grant #231575 Open Philanthropy 182 Howard Street #225 San Francisco, CA 94105 https://www.openphilanthropy.org/ Open Philanthropy played no role in the study design, data collection and analysis, decision to publish, or preparation of the manuscript.

**Competing interests:** The authors have declared that no competing interests exist.

population's average Healthy Eating Index (HEI) score, from 56 in 2005–6 to 59 in 2015–6 (out of a maximum score of 100). However, because diet quality has been found to be higher for U.S. adults who eat away from home less frequently [6], and because restaurant sales have been shown to decline by one-third during the pandemic [1], there is a possibility that the pandemic-induced decrease in restaurant sales could improve overall diet quality during the early days of the pandemic.

Early studies investigating the pandemic's impact on food behavior and diet quality find mixed results: A study of British households finds that the decrease in calories purchased from dine-in restaurants was more than made up for by an increase in calories purchased from grocery stores and to-go restaurants [7]. This same study, which uses supermarket scanner data covering purchase for at-home consumption as well as a novel dataset that records purchases for out-of-home consumption, also finds a small decrease in overall diet quality in the first two-to-three months of the pandemic.

A systematic review [8] of pandemic-related food behavior studies summarizes five studies that focused on North America. Of these, one focused on Canada and shows an increase in diet quality as measured by the 2015 HEI [9]. The four focused on the U.S. generally find little change in overall diet quality, as measured by a variety of indicators ranging from dietary recall for specific types of foods consumed [10, 11], dietary recall on amounts of added sugars consumed [12], and agreement with a survey statement about changes in healthy food consumption [13]. For example, one of these studies finds an increase in sweets and salty snacks but also increases in fruits, eggs, and poultry, and non-starchy vegetables, and another finds an increase in fresh produce, dairy, grains, and a decrease in fast food and meat [10, 11]. Adding to this literature, the current study has two salient features: First, it uses household-level purchase data from the NielsenIQ consumer panel, sometimes called "scanner data". Second, it focuses on two practical and easy-to-measure indicators of diet quality. The first indicator is a diet diversity index, also known as a *Berry Index* [14], which ranges from 0 (least diverse) to 1 (most diverse). Numerous studies show that higher food group variety correlates with healthier outcomes [5, 15, 16]. As shown in the Methods section, our construction of the *Berry Index* relies on household-level expenditure shares for the 24 food categories used by the Center for Nutrition Policy and Promotion (CNPP). The second indicator, called *USDAScore*, has been developed by USDA to measure diet quality [17] and subsequently used as a measure of diet quality in several other studies [18–20]. Our construction of the *USDAScore* likewise uses household-level expenditure shares for the 24 CNPP food categories, where each share is the portion of category expenditure relative to total household food purchase expenditures. However, unlike the *Berry Index*, each household's expenditure share is compared to "recommended" expenditure shares based on USDA's Thrifty Food Plan. The *USDAScore* penalizes deviations from the recommended shares, and it increases as household expenditures more closely match USDA recommended shares. These two scores, the *Berry Index* and the *USDAScore*, are easily calculated from household-level scanner data without the need for product-level nutritional information, which is required for calculation of the HEI [21].

Using our two measures of dietary quality, we construct an event-study difference-in-differences where the pandemic-induced event is based on the date when U.S. schools were closed, which ranged from March 12, 2020 (Michigan) to April 2, 2020 (Maine). Because the dates of closure vary by state, we rescale the data to a time-since-closure scale so that each household's closure week is labeled as "week 0". We then construct, for each household, our two diet quality measures for a set of four-week periods (i.e., approximately one month) extending six months before (leads) and after (lags) the date of closure in each household's state. To address seasonality in diets and purchasing behavior and to construct an appropriate counterfactual,

each four-week diet quality calculation is compared against a calculation for the same household exactly one year earlier.

## Materials and methods

### Data

Household grocery purchases and characteristics are obtained from the NielsenIQ Homescan Consumer Panel data that provide detailed trip-level food purchases from a nationally representative panel of over 60,000 U.S. households. Recording quantity and price paid for every UPC purchased, this data allow the study to separate purchases of different food product categories. Our sample consists of households who were panelists in the Consumer Panel dataset from late August 2018 through September 2020. Households who migrated to a different county during the period are dropped from the panel to minimize confounding factors due to relocation and adaptation to a new food and geographical environment. After applying these two filters, 41,579 households remain in our dataset.

### Event study model

We use an event-study approach to identify the causal relationship between the pandemic event and changes in food-purchasing behavior. Because not all U.S. states responded to the pandemic simultaneously, for each household the pandemic event is defined as the start of school closures recommended by the state government of a household's county of residence in 2020. Although the recommended start of school closures in each state is identifiable by a calendar date, this study defines the event as a 4-week "event month" or "school closure month" whose exact weeks of the year may vary by state but are identical for all households residing within the same state. Each week is defined as a Sunday-to-Saturday cycle. The event month, which spans the school closure date, then becomes the reference month ($m = 0$) in all event-study figures, and all other months–leads and lags–in the sample period are relative to the event month.

The study period ranges from 6 months before to 6 months after the event month (school-closure month) in each state. Consistent with the event month, each relative "month" is defined as a period of 4 consecutive weeks following methods from a recent study [7]. The monthly aggregation allows this study to observe more meaningful changes, especially regarding grocery composition, amidst possibly less frequent shopping by households. Effectively, the pandemic year in this study consists of 52 weeks (collapsed into 13 relative months that each contains 4 weeks) whose range may slightly vary by state depending on the week of the first day of school closure. Aggregating household purchases to relative month-level observations also reduces the number of "zero observations" in the sample because not all households shop for groceries every week.

To control for the average seasonal trend of household purchases across relative months, this study assembles a set of "control year" observations using observations from 52-weeks prior to each observation during the pandemic year. Therefore, this study uses each household's own food purchases during the control year to inform the counterfactual.

The basic regression model in this study is based on the following equation:

$$FoodOutcome_{i,m,y}$$
$$= \sum_m \beta_m (PandemicYear_i \; x \; RelativeMonth_m) + \gamma PandemicYear_i + \sum_m \delta_m RelativeMonth_m$$
$$+ \; \varepsilon_{i,m,y}, (1)$$

where households are indexed by $i$, relative months are indexed by $m$ (-6 $\leq m \leq$ 6), and relative years are indexed by $y$. The food diversity and healthfulness scores, defined next, comprise the dependent variables. The $\delta_m$ coefficients capture relative month $m$ fixed effects. *Pandemic-Year* is an indicator variable which takes the value of 1 if the observation is during the pandemic year and 0 if during the control year. In Eq (1), each $\beta$ coefficient carries the interpretation of average change in household food outcomes for a given month in the pandemic year compared to the same month in the previous year (2019). Note that many other event-study models can accomplish this same comparison using a rich set of household and month fixed effects; however, since our relative months do not accurately correspond to calendar months, this relatively standard method is not available.

In the event-study plots, Eq (1) is estimated for each sample or subsample, and the recovered estimates of the $\beta$ coefficients from these regressions are plotted. These estimated coefficients capture average year-to-year shifts in household food choices. To establish the causality that the early pandemic affected household food choices, the event-study plots must show clear and significant disturbances in trends compared to trends of the pre-event months.

**Food diversity score.** While studies investigating food diversity vary slightly in the precise grouping of foods, most measures start with a similar functional form applied by Berry [14]. The *Berry Index*, also known as the Simpson Index, is defined as:

$$BI_{imy} = 1 - \sum_c s_{icmy}{}^2 \qquad (2)$$

where $s_c$ represents the share of food category $c$ for each $i$ household and $m$ month. The *Berry Index* can therefore take a value between 0 (least diverse) and 1 (most diverse), and in this raw form, it is also considered a measure of "evenness" of composition where scores near 0 indicate the most uneven composition and scores near 1 indicate nearly equal shares. Traditionally, studies use food intake data from questionnaires or food diaries for a cross-section of individuals and the shares are based on the tally of food groups, subgroups, or unique items (divided by the total number) that are consumed by the individual during the observation period. However, instead of tallying the number of unique food groups, this study employs shares of 24 food categories based on actual household food-purchase expenditures from NielsenIQ Homescan data. The 24 food categories follow the definition by the CNPP and are shown in Table 1. Hence, for each household-month-year observation, this study calculates the food expenditure share for each category to then calculate the respective *Berry Index*.

**Food healthfulness score.** To better understand whether American household food spending compositions are moving towards diet guidelines, USDA researchers have created a *USDAScore* that accounts for recommended expenditure shares for various food groups as points of comparison against a household's actual expenditure shares [14]. Recommended shares are based on the USDA's Thrifty Food Plan that is designed to meet the requirements of the USDA's recommended healthy diet according to the Dietary Guidelines for Americans [22]. Having further aggregated food categories from the Quarterly Food at Home Price Database (QFAHPD) to CNPP categories and matching them with the weekly food costs [23], USDA researchers effectively calculate the USDA Score as a squared-error loss function which aggregates the deviations between the recommended food expenditure shares and the actual expenditure shares across 24 food categories [17]. Using the first version of the *USDAScore*, this study defines the *USDAScore* for each household $i$ by the following formula:

$$USDAScore_{imy} = \left( \sum_c \left( expshare_{icmy} - USDAexpshare_{icmy} \right)^2 \right)^{-1} \qquad (3)$$

where $i$ indexes individual households, $c$ denotes the CNPP food categories, $m$ indexes the

**Table 1. CNPP food categories, average recommended expenditure shares, and average actual expenditure shares for the full sample over the entire study period.**

| Broad category | Category code (in this study) | CNPP food category | Average recommended expenditure share in household sample (%) | Average actual "monthly" expenditure share in household sample (%) |
|---|---|---|---|---|
| Grains | 1 | Whole grains | 10.11 | 0.64 |
| | 2 | Non-whole grains | 4.44 | 25.49 |
| Vegetables | 3 | Potatoes | 2.07 | 1.04 |
| | 4 | Green vegetables | 8.48 | 0.77 |
| | 5 | Orange vegetables | 2.13 | 1.32 |
| | 6 | Legumes | 8.04 | 0.98 |
| | 7 | Other vegetables | 8.98 | 4.44 |
| Fruits | 8 | Whole fruits | 15.94 | 5.72 |
| | 9 | Juices | 1.24 | 2.39 |
| Milk products | 10 | Whole milk yogurt | 1.15 | 7.33 |
| | 11 | Non-whole milk yogurt | 10.38 | 1.01 |
| | 12 | Cheese | 0.39 | 4.72 |
| Meat and beans | 13 | Meats | 6.18 | 5.59 |
| | 14 | Poultry | 3.81 | 1.94 |
| | 15 | Fish | 8.55 | 2.05 |
| | 16 | Processed meats | 0.51 | 4.83 |
| | 17 | Nuts | 3.64 | 1.85 |
| | 18 | Eggs | 0.15 | 0.97 |
| Other foods | 19 | Condiments | 1.48 | 4.83 |
| | 20 | Coffee tea | 0.05 | 3.43 |
| | 21 | Soft drinks | 0.86 | 5.50 |
| | 22 | Sweets | 0.31 | 5.70 |
| | 23 | Soups | 1.05 | 1.29 |
| | 24 | Entrees | 0.05 | 6.18 |

relative month, and *y* indexes the relative year (pandemic year or control year). While the variable *expshare* is the food expenditure share calculated from the actual household purchase data, *USDAexpshare* for each category *c* is the recommended shares and is constant across households, months, and years. As a measure of healthfulness, a higher *USDAScore* indicates a more healthful mix of food purchases. For the 41,579 households in our sample, Table 1 presents the actual and recommended expenditure shares for all households averaged over the entire study period that runs from late August 2018 to mid-September 2020.ased on Eq (3), this study calculates a *USDAScore* for each household-month-year observation. Since the product descriptions in the recent NielsenIQ data do not directly map to the CNPP food categories, the 816 product module codes of foods–from dry grocery, deli, fresh produce, frozen foods, packaged meat departments, as well as food magnet data–are first manually mapped to the 24 CNPP categories. This effort builds on a prior study with the NielsenIQ data that also required aggregating over 600 broad NielsenIQ food categories available during that period into 52 food groups for the QFAHPD [24]. While some grouping definitions may slightly vary across different studies, the results of this study are less likely to be sensitive to the groupings since the main coefficients of interest in the regression Eq (1) represent average changes in food healthfulness levels instead of the average levels themselves.

**Data analysis.** Raw data of school closure dates by state are manually entered into Microsoft Excel file and imported into STATA (Version 17.0). Raw NielsenIQ data in Globus have

**Table 2. Summary statistics of household sample food outcomes.**

| Outcome Variable | Relative year | Mean | SD | Min | 25th percentile | 50th percentile | 75th percentile | Max |
|---|---|---|---|---|---|---|---|---|
| Food diversity score | *Control year* | 0.82 | 0.13 | 0.0 | 0.80 | 0.86 | 0.89 | 1.00 |
| *Berry Index—24 categories* | *Pandemic year* | 0.82 | 0.13 | 0.0 | 0.81 | 0.86 | 0.89 | 1.00 |
| Food healthfulness score | *Control year* | 6.45 | 2.82 | 0.9 | 4.48 | 6.21 | 8.10 | 35.89 |
| *USDAScore* | *Pandemic year* | 6.54 | 2.83 | 0.9 | 4.59 | 6.31 | 8.18 | 36.42 |

been downloaded and imported into STATA (Version 17.0) for merging with previously imported data and analyses. Regressions are estimated using the *reghdfe* package and the results are exported to Microsoft Excel for creating figures.

## Results

For the 41,579 households in our sample, Table 2 compares the average *Berry Index* and *USDAScore* for the entire study period that runs from late August 2018 to mid-September 2020, i.e., the pandemic year, and the prior year. The means and various quantiles are very similar, a result that is partially expected because half of the calculations from 2020 are in the pre-school closure/pre-pandemic period.

Table 3 presents the mean values for the *Berry Index* and *USDAScore* by relative month. For relative months –6 to –1, the table shows that the *Berry Index* and *USDAScore* for the control year and pandemic year are close in value, with some months being lower and others higher. However, for relative months 0 to 6, the *Berry Index* and *USDAScore* are always higher in the pandemic year.

Fig 1 presents the estimates of year-to-year changes in the *Berry Index* and *USDAScore* where each household's monthly score is relative to their score in the previous year. Specifically, the graphed values reflect monthly parameter estimates from two regressions using the *Berry Index* and *USDAScore* as dependent variables. Full regression results are available in the S1 Table. Fig 1 shows a 1.4 to 2.6 percent increase for the *Berry Index* and a 6.6 to 8.5 percent increase for the *USDAScore* in the event month and subsequent two months. The *Berry Index*

**Table 3. Mean values of *Berry Index* and *USDAScore* by relative month.**

| Relative month | Berry Index | | USDA Score | |
|---|---|---|---|---|
| | Control year | Pandemic year | Control year | Pandemic year |
| -6 | 0.8195 | 0.8144 | 6.3583 | 6.2879 |
| -5 | 0.8234 | 0.8190 | 6.4355 | 6.3281 |
| -4 | 0.8277 | 0.8244 | 6.5448 | 6.4942 |
| -3 | 0.8254 | 0.8211 | 6.4710 | 6.3473 |
| -2 | 0.8260 | 0.8224 | 6.6113 | 6.5138 |
| -1 | 0.8214 | 0.8192 | 6.5306 | 6.4355 |
| 0 | 0.8195 | 0.8351 | 6.5127 | 6.8474 |
| 1 | 0.8195 | 0.8309 | 6.4731 | 6.7938 |
| 2 | 0.8141 | 0.8287 | 6.3983 | 6.8206 |
| 3 | 0.8133 | 0.8225 | 6.4328 | 6.6746 |
| 4 | 0.8147 | 0.8211 | 6.4862 | 6.6248 |
| 5 | 0.8112 | 0.8183 | 6.3282 | 6.4993 |
| 6 | 0.8133 | 0.8181 | 6.3061 | 6.3980 |

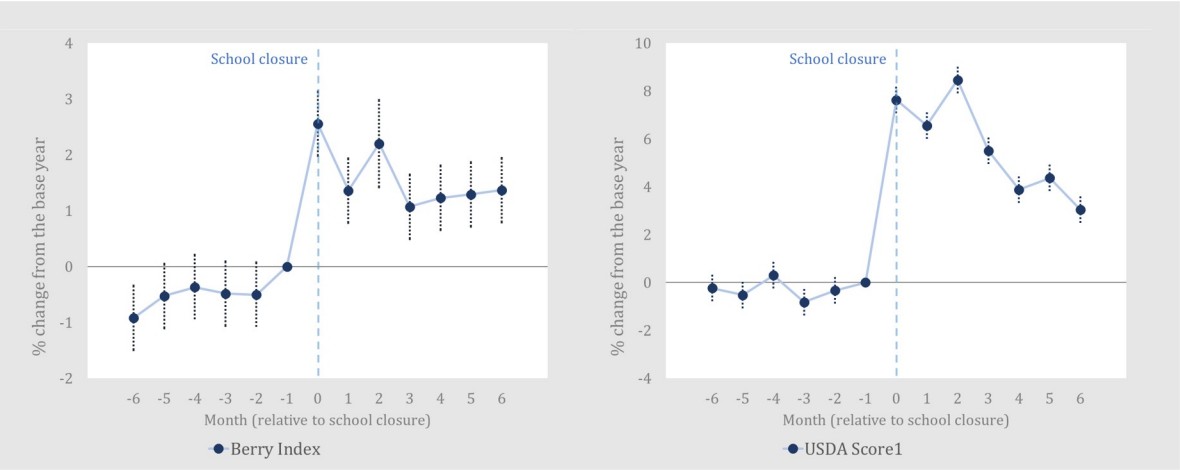

**Fig 1. Year-on-year household-level changes in the *Berry Index* (Diet diversity) and *USDAScore* (Diet healthfulness), pre- and post-pandemic.** This is constructed from regression results in S1 Table and the vertical bars represent 95% confidence intervals. The proportional increases on the vertical axes can be easily converted to % increases. Thus, a value of 0.02 or 0.04, as examples, is equivalent to a 2% or 4% increase. The values can be interpreted as year-on-year changes due to the pandemic.

then drops to a 1.2 percent increase in the third month after school closures and then increases slightly throughout the rest of the study period. On the other hand, the *USDAScore*'s increase diminishes each month after its peak in the second month after school closures.

In Fig 2, we generally find that the *Berry Index* jumps more than 2 percent (approximately) and then levels off for most subsamples of households based on a range of characteristics. That said, household characteristics do affect the trends in the *Berry Index*, and Fig 2 suggests some important results: (a) Midwestern households have larger *Berry Index* increases than those from other regions; (b) Households with young children or a mix of school-age children have lower increases in the first couple of months after school closures than households with no children or households with older children; (c) low-income households (defined here as households earning less than $30,000 per year) have a noticeable decline in the Berry Index increases 2 to 5 months after school closures; (d) Households classified as Asian ethnicity or Other ethnicity generally have higher increases; (e) Households earning no income (which can include both retirees and unemployed household heads) saw no significant increase in the *Berry Index* one month past school closures and generally low increases in the following months; and (iv) Households that do not own a vehicle (a classification assigned to households that never bought any items in the "automotive" category at least recent two years of NielsenIQ data) have substantially lower increases than those that own one (i.e., households that purchased automotive items in recent years).

Fig 3 repeats some of these findings for the *USDAScore*. Households with young children have substantially lower increases in the *USDAScore* (in early months). Low-income households have substantially smaller increases in the *USDAScore* than higher income households. In fact, the *USDAScore* increases by over 10 percent for the highest income households in the second month after the school closures. Households classified as White ethnicity have lower increases, while households classified as Asian ethicity have the highest, with the *USDAScore* increasing by almost 15 percent in the second month. Finally, households that do not own a vehicle have slightly lower increases in months 1 and 2. Summary statistics for all household demographic variable used in Figs 2 and 3 are found in S3 Appendix.

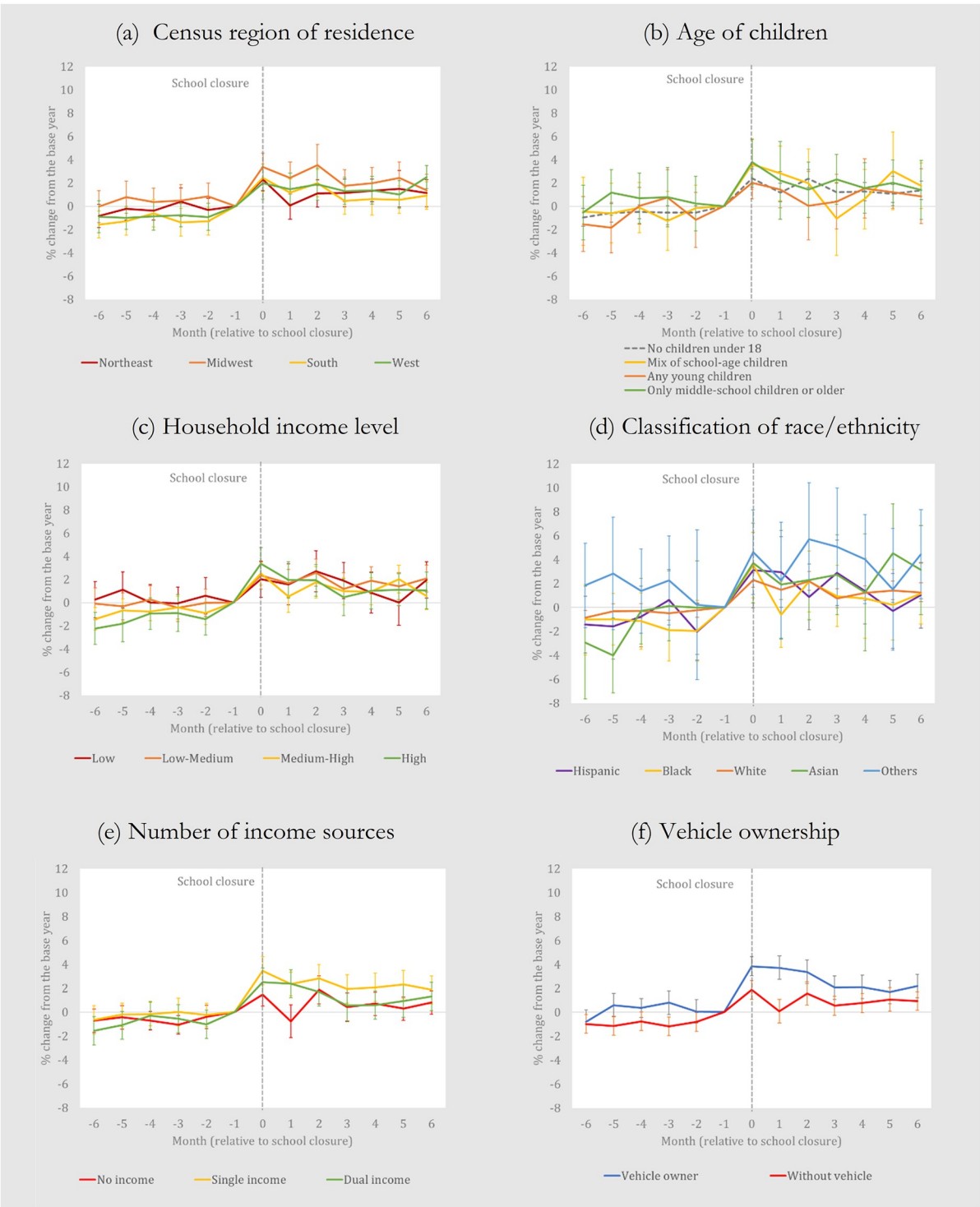

**Fig 2. Year-on-year household-level changes in the *Berry Index* (Diet diversity), pre- and post-pandemic, by household characteristic.** This is constructed from regression results in S1 Appendix and the vertical bars represent 95% confidence intervals. The proportional increases on the vertical axes can be easily converted to % increases. Thus, a value of 0.02 or 0.04, as examples, is equivalent to a 2% or 4% increase. The values can be interpreted as year-on-year changes due to the pandemic.

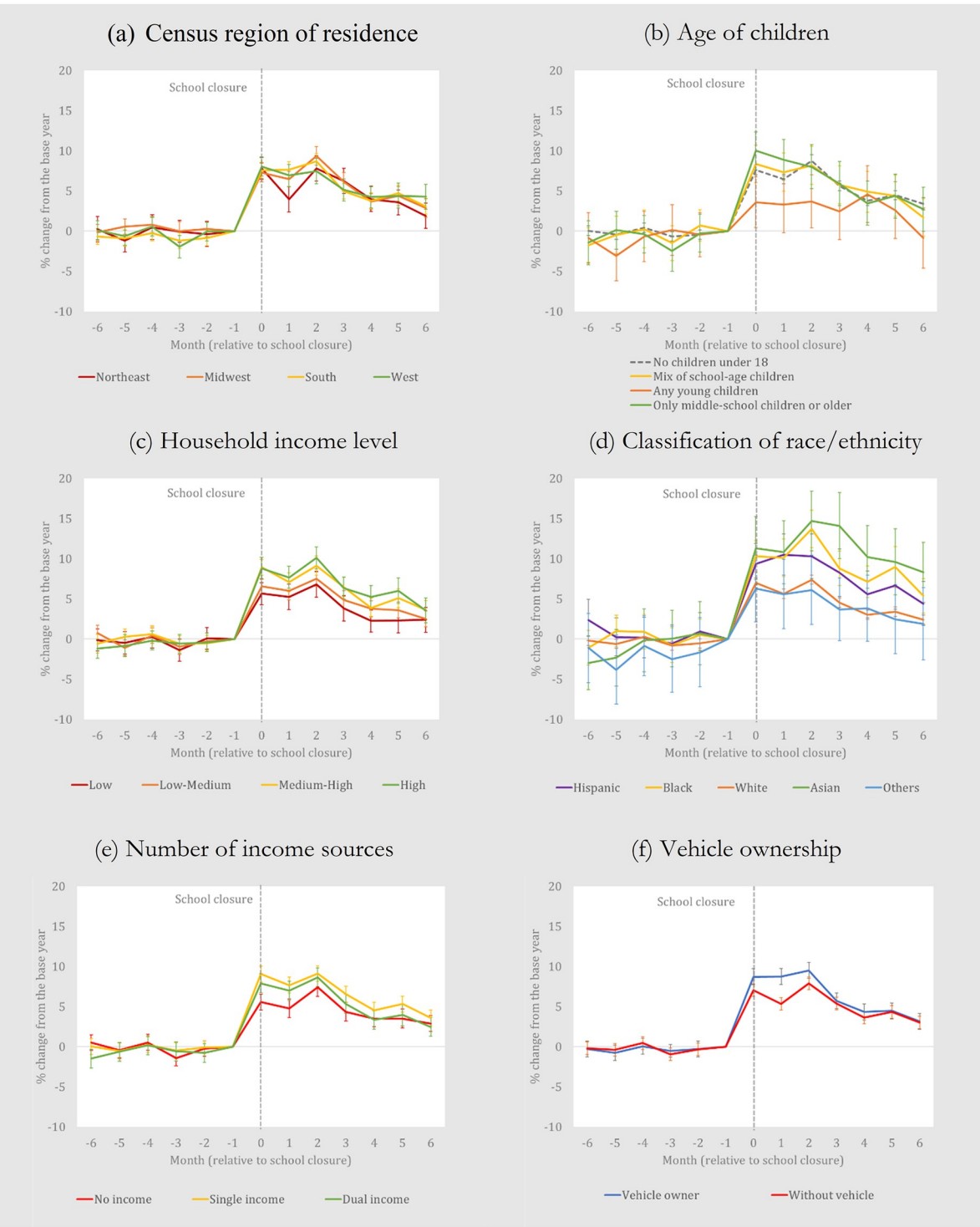

**Fig 3. Year-on-year household-level changes in the *USDAScore* (Diet healthfulness), pre- and post-pandemic, by household characteristic.** This is constructed from regression results in S2 Appendix and the vertical bars represent 95% confidence intervals. The proportional increases on the vertical axes can be easily converted to % increases. Thus, a value of 0.02 or 0.04, as examples, is equivalent to a 2% or 4% increase. The values can be interpreted as year-on-year changes due to the pandemic.

## Discussion

Like prior research, our results confirm that the COVID-19 pandemic dramatically affected the food-purchasing behavior of U.S. households. However, unlike other studies, we find modest temporary increases in food diversity (of up to 2.6 percent compared to the prior year) as measured by the *Berry Index* and larger temporary increases in diet quality (of up to 8.5 percent) as measured by the *USDAScore*. To put these results in context, no study that we know of investigates pandemic-induced changes in food diversity and very few studies investigate diet quality changes in the U.S. market. One of the few studies that does find an increase in diet quality comes from questionnaire data of Canadian households, and contrary to the authors' prior expectations, but like our results, finds an overall increase in diet quality as measured by the HEI-2015 [9]. A study of British households used scanner data, like our study, but found little change in overall diet quality as measured by the HEI [7].

Readers should recall, however, that our results are based only on food-at-home purchase data found in the NielsenIQ Homescan dataset. Therefore, the increases shown in Figs 1–3 might not accurately reflect comprehensive food diversity or food healthiness changes if FAFH data were also included. In other words, it is hypothetically possible that food diversity or healthfulness scores calculated using both food-at-home and FAFH data might not show the same increases. A study of British households uses data from two sources, one for at-home purchases and one for out-of-home purchases, and finds a slight decline in diet quality in the early months of the crisis [7]. While it is unavailable for our study, incorporating out-of-home data to our event study could alter our results.

A second caveat involves our diet quality score, the *USDAScore*. In the months following pandemic closures, we find that the *USDAScore* shows substantial increases (i.e., more than the *Berry Index*). While many research studies addressing diet quality now use the HEI, its use requires substantially more data that link individual food purchases to nutritional content. Thus, we show that the *USDAScore* substantially increases, but since we are unable to investigate the HEI score with our dataset, we do not know if the HEI would also show substantial increases. It would be useful for researchers with access to HEI-required data to confirm the healthfulness increase.

A third set of caveats centers on possible inaccuracies of the *Berry Index* and *USDAScore* due to potential stockpiling by households and food price inflation during the pandemic. Studies [25, 26] based on consumer surveys during the early part of the school-closure period do find evidence of stockpiling (which varies across household type). However, because our study period continues for six months past the closure week, and because we examine four-week "months" rather than weekly purchases, we expect panic-induced stockpiling may play only a small role in our results. While food inflation could lead to additional inaccuracies with our dietary measures, the fact that the *Berry Index* and *USDAScore* are based on a food category's expenditure share rather than expenditure level, helps to mitigate this concern. In other words, if all food group prices increase proportionally, then our two measures are unaffected by inflation. However, some product groups, such as eggs and meats, may have experienced higher rates of price inflation than other groups. Table 1 shows that meats have a high average expenditure share, so an inflation-induced increase in this share would lead to a lower *Berry Index*, contrary to what we observe. On the other hand, eggs have a low average expenditure share, so an inflation-induced increase in this share would lead to a higher *Berry Index*. Our hope is that these potential inflation-related inaccuracies may cancel each other out over the full set of 24 food categories.

These three caveats aside, there are several possible household-level mechanisms for pandemic-induced increases in food diversity and healthfulness:

First, since other studies [6] do find food from dine-in restaurants and other FAFH outlets to be less healthy than food-at-home purchases from supermarkets and other food retailers, it is possible that the dramatic decrease in restaurant purchases [1] positively affected food-at-home food diversity and healthfulness. That possibility suggests that any future shifts away from restaurant expenditures, even those not caused by the pandemic, could improve food diversity and healthfulness.

Second, it is also likely that the pandemic increased consumers' focus on health and caused a shift in food-at-home shopping towards more diverse and healthy foods for some but not all households. Several research studies support this notion of greater health consciousness during the pandemic [27–29]; however, these studies are not set in the U.S., so we have no direct evidence that health consciousness increased there.

Third, it is known that the pandemic caused some stockouts at grocery stores and supermarkets, and it is likely that food-at-home consumers shifted to new foods that led to increased diversity and healthfulness. Related research on natural disasters finds that hurricanes in the U.S. cause substantial brand switching in the bottled water category, presumably because their usual brand was not available [30]. It is likely that during the COVID-19 pandemic, unavailable product categories may have caused similar switching behaviors, but on a broader scale. Our results suggest that at least some of this switching and experimentation with new foods contributed to increases in diversity and healthier food choices that did not completely dissipate by the end of the study period.

Lastly, pandemic-induced school and business closures may have altered time constraints within many households, with some households having more time to cook and prepare foods and others, such as those with young children, having less time. In addition, if some of these same households maintained their same employment levels yet faced lower costs (via savings in transportation, childcare, and/or recreation) then disposable income could actually increase in some cases. Thus, for households with less-binding time constraints due to pandemic closures, and for households with both less-binding time constraints and higher disposable income, diet diversity and healthfulness are likely to improve. Indeed, some existing studies point to this outcome: One study of British households finds a shift in the composition of calories purchased toward raw ingredients used for cooking, which the authors say is consistent with a decrease in the opportunity cost of time during the pandemic [7]. A systematic review of COVID-related research suggests that, overall, families spent more time planning meals, cooking, and eating together [31]. Separately, other research shows that cooking at home is associated with better diet quality [32–34].

## Conclusions

Despite some supply chain problems (or maybe because of it), we find that household-level food diversity and healthfulness increased during the six months after pandemic-induced school closures. Food diversity as measured by the *Berry Index* rose as much as 2.6 percent in the first few months of the school-closure period, on average, followed by small increases and a leveling off. Food healthfulness, as measured by the *USDAScore*, increased by as much as 8.5 percent, on average, in the first months of the closure period, again followed by a gradual decrease. Six months after closures, both scores were still above normal levels (as measured by food purchasing patterns by the same households one year prior). Finally, households with different demographic characteristics generally follow the same patters; however, some groups (such as households with young children, low-income households, and households that do not own a vehicle) exhibit smaller increases.

## Supporting information

**S1 Table. Results for *Berry Index* and *USDAScore* regressions.**
(DOCX)

**S1 Appendix. Results for *Berry Index* regressions with heterogeneity.**
(DOCX)

**S2 Appendix. Results for *USDAScore* regressions with heterogeneity.**
(DOCX)

**S3 Appendix. Definition of household demographic variables and summary statistics.**
(PDF)

## Acknowledgments

The authors also acknowledge the Kilts Center for Marketing and NielsenIQ. The NielsenIQ Homescan Consumer Panel is available via a subscription fee from the Kilts Center for Marketing as part of the University of Chicago Booth School of Business (https://www.chicagobooth.edu/research/kilts). The data are confidential and cannot be shared, and research projects must be registered with the Kilts Center.

Researcher(s)' own analyses calculated (or derived) based in part on data from NielsenIQ Consumer LLC and marketing databases provided through the NielsenIQ Datasets at the Kilts Center for Marketing Data Center at The University of Chicago Booth School of Business.

The conclusions drawn from the NielsenIQ data are those of the researcher(s) and do not reflect the views of NielsenIQ. NielsenIQ is not responsible for, had no role in, and was not involved in analyzing and preparing the results reported herein.

## Author Contributions

**Conceptualization:** Edward C. Jaenicke, Douglas H. Wrenn.

**Data curation:** Daniel P. Simandjuntak.

**Formal analysis:** Daniel P. Simandjuntak.

**Funding acquisition:** Edward C. Jaenicke.

**Methodology:** Douglas H. Wrenn.

**Project administration:** Edward C. Jaenicke.

**Supervision:** Edward C. Jaenicke, Douglas H. Wrenn.

**Writing – original draft:** Daniel P. Simandjuntak.

**Writing – review & editing:** Edward C. Jaenicke, Douglas H. Wrenn.

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
