## [Decision Letter · Decision Letter 0]

19 Dec 2023

PONE-D-23-26121Pandemic-induced changes in household-level food diversity and diet quality in the U.S.PLOS ONE

Dear Dr. Jaenicke,

Thank you for submitting your manuscript to PLOS ONE. After careful consideration, we feel that it has merit but does not fully meet PLOS ONE’s publication criteria as it currently stands. Therefore, we invite you to submit a revised version of the manuscript that addresses the points raised during the review process. Please submit your revised manuscript by Feb 02 2024 11:59PM. If you will need more time than this to complete your revisions, please reply to this message or contact the journal office at plosone@plos.org. Please include the following items when submitting your revised manuscript:A rebuttal letter that responds to each point raised by the academic editor and reviewer(s). You should upload this letter as a separate file labeled 'Response to Reviewers'.A marked-up copy of your manuscript that highlights changes made to the original version. You should upload this as a separate file labeled 'Revised Manuscript with Track Changes'.An unmarked version of your revised paper without tracked changes. You should upload this as a separate file labeled 'Manuscript'.

We look forward to receiving your revised manuscript.

Kind regards,

Zhifeng Gao

Academic Editor

PLOS ONE

Journal Requirements:

Reviewers' comments:

Reviewer's Responses to Questions

**Comments to the Author**

1. Is the manuscript technically sound, and do the data support the conclusions?

Reviewer #1: Yes

Reviewer #2: Yes

2. Has the statistical analysis been performed appropriately and rigorously? 

Reviewer #1: Yes

Reviewer #2: Yes

3. Have the authors made all data underlying the findings in their manuscript fully available?

Reviewer #1: Yes

Reviewer #2: Yes

4. Is the manuscript presented in an intelligible fashion and written in standard English?

Reviewer #1: Yes

Reviewer #2: Yes

5. Review Comments to the Author

Reviewer #1: Referee report for: PONE-D-23-26121

Summary: This manuscript assesses changes in household dietary diversity using two different measures during the year following the COVID-19 pandemic. The manuscript uses an event study design with a pre-pandemic control group constructed based on the beginning of school lockdowns at the state level. Additionally, the study uses scanner data to construct the dietary diversity measures. Overall, the study finds that the COVID-19 lockdowns were associated with increases in the dietary diversity score (across both measures). The manuscripts proceed to discuss several mechanisms for this increase including the decreased availability of food away from home, changes in health consciousness, supply chain stockouts, and changing time constraints.

• In the conclusion and discussion section, the proposed mechanism of decreased FAFH availability seems outside the scope of this study. As correctly noted, the present measures do not include any FAFH purchases and the study provides no evidence on the prevalence of restaurant closures or FAFH availability during these periods across states. Anecdotally, FAFH available for takeout and delivery appears to have been widely available during this period. Also, pre-prepared foods at grocery retailers may have similar health attributes as FAFH and could have been included in these grocery baskets. Again, I think this mechanism is somewhat of a stretch given the limits of the data presented in this article.

• Why restrict the time period? Is it possible to extend the comparison and keep the control category (i.e. correct reference month before the pandemic) the same?

• All of the figures need vertical axis labels.

• In Table 2, is it possible to calculate the mean scores by month for control and pandemic? This might help provide additional intuition to support the regression estimates especially since the differences between annual averages don’t look meaningful. And by month I mean the recoded 4 week periods constructed in the analysis, not necessarily calendar months.

• It would be helpful if the article could do more to discuss the real significance of the measured change. For example, what might a 2 percent change in the Berry Index look like? Is that just consumption of one more product or a fraction of a product?

• Relatedly, from construction of the indices can the authors assess where the increased diversity came from? Even just looking at the broad categories might be instructive. For example, given the supply and income shocks motivating the manuscript we might think diversity declined due to stock outs and limited income. However, given the finding of increasing diversity were consumers purchasing additional fruits/vegetables or something else?

• More details on the construction of the alternate demographic variables would be helpful. For example, what income levels were considered? Why is single income grouped with no income? Summary statistics for the sample and definitions for what the actual survey question looked like would be helpful.

• Are the authors able to add any discussion of income effects that may have also been at play during the COVID-19 pandemic? This is especially important given the great deal of research supporting dietary diversity changes with income. Also it appears there is a sharper decline for the low income category in Figure 2 in the months following the pandemic.

The manuscript references supply chain shocks, but at the same time we know there were income shocks which also occurred both from lost employment and from government distribution of payments. For example, Lai et al. (2020) find that over 65% of their sample had received an Economic Impact Payment by mid-May 2020 which covers the early period of most of your sample and may have influenced purchasing or consumption behavior. Below are two references which I think may be helpful at framing this discussion that look at consumer purchasing and expenditures: Bina et al. (2023) and Lai et al. (2020).

References:

Bina, J. D., Tonsor, G. T., & Briggeman, B. C. (2023). COVID-19 Federal Aid and Household Food Expenditures. Journal of Agricultural and Applied Economics, 55(4), 567-608.

Lai, J., Morgan, S., Kassas, B., Kropp, J., & Gao, Z. (2020). Spending of economic stimulus payments and changes in food purchasing during the COVID-19 pandemic. Choices, 35(3), 1-8.

Reviewer #2: In general, I find this paper to be well-written and consider its research topic to be of significant importance to the field. Understanding consumers’ diet change caused by the pandemic holds substantial implications for developing food policy to increase US consumers’ healthy food consumption. Nevertheless, I have some concerns in this paper, which I have detailed in the attachment.

6. PLOS authors have the option to publish the peer review history of their article (what does this mean?). If published, this will include your full peer review and any attached files.

Reviewer #1: No

Reviewer #2: No

---

## [Author Response · Author response to Decision Letter 0]

30 Jan 2024

Our response to reviewers has been uploaded in a PDF file titled "Response to Reviewers."

---

## [Decision Letter · Decision Letter 1]

6 Mar 2024

Pandemic-induced changes in household-level food diversity and diet quality in the U.S.

PONE-D-23-26121R1

Dear Dr. Jaenicke,

We’re pleased to inform you that your manuscript has been judged scientifically suitable for publication and will be formally accepted for publication once it meets all outstanding technical requirements.

Kind regards,

Zhifeng Gao

Academic Editor

PLOS ONE

Additional Editor Comments (optional):

Please check reviewer's comment to make minor changes. Particularly, the comment, "However, there is a missing number in the abstract. It would be great if you can change "XXX" to a real number."

Reviewers' comments:

Reviewer's Responses to Questions

**Comments to the Author**

1. If the authors have adequately addressed your comments raised in a previous round of review and you feel that this manuscript is now acceptable for publication, you may indicate that here to bypass the “Comments to the Author” section, enter your conflict of interest statement in the “Confidential to Editor” section, and submit your "Accept" recommendation.

Reviewer #1: All comments have been addressed

Reviewer #2: All comments have been addressed

2. Is the manuscript technically sound, and do the data support the conclusions?

Reviewer #1: Yes

Reviewer #2: Yes

3. Has the statistical analysis been performed appropriately and rigorously? 

Reviewer #1: Yes

Reviewer #2: Yes

4. Have the authors made all data underlying the findings in their manuscript fully available?

Reviewer #1: No

Reviewer #2: No

5. Is the manuscript presented in an intelligible fashion and written in standard English?

Reviewer #1: Yes

Reviewer #2: Yes

6. Review Comments to the Author

Reviewer #1: Thank you for your thoughtful response to comments. I do believe that the inclusion of RR Tables 1 and 2 would improve the manuscript by providing some information about the changes (and the limits of our understanding) in these indices over time.

Reviewer #2: I would like to applaud and thank the authors for adequately addressing my comments and other reviewers' comments. I enjoy reviewing this paper. However, there is a missing number in the abstract. It would be great if you can change "XXX" to a real number.

7. PLOS authors have the option to publish the peer review history of their article (what does this mean?). If published, this will include your full peer review and any attached files.

Reviewer #1: No

Reviewer #2: No
